# The Influence of the Structure of Selected Polymers on Their Properties and Food-Related Applications

**DOI:** 10.3390/polym14101962

**Published:** 2022-05-11

**Authors:** Piotr Koczoń, Heidi Josefsson, Sylwia Michorowska, Katarzyna Tarnowska, Dorota Kowalska, Bartłomiej J. Bartyzel, Tomasz Niemiec, Edyta Lipińska, Eliza Gruczyńska-Sękowska

**Affiliations:** 1Department of Chemistry, Institute of Food Sciences, Warsaw University of Life Sciences, 02-776 Warsaw, Poland; piotr_koczon@sggw.edu.pl (P.K.); katarzyna_tarnowska@sggw.edu.pl (K.T.); dorota_kowalska@sggw.edu.pl (D.K.); 2Independent Researcher, 02-091 Warsaw, Poland; heidi.i.josefsson@gmail.com; 3Department of Bioanalysis and Drug Analysis, Faculty of Pharmacy, Medical University of Warsaw, 02-097 Warsaw, Poland; ssolobodowska@wum.edu.pl; 4Department of Morphological Sciences, Institute of Veterinary Medicine, Warsaw University of Life Sciences, 02-776 Warsaw, Poland; bartlomiej_bartyzel@sggw.edu.pl; 5Animals Nutrition Department, Institute of Animal Sciences, Warsaw University of Life Sciences, 02-786 Warsaw, Poland; tomasz_niemiec@sggw.edu.pl; 6Department of Biotechnology, Microbiology and Food Evaluation, Institute of Food Sciences, Warsaw University of Life Sciences, 02-776 Warsaw, Poland; edyta_lipinska@sggw.edu.pl

**Keywords:** biopolymer, polysaccharide, food-related applications, food industry, enzyme immobilisation, starch

## Abstract

Every application of a substance results from the macroscopic property of the substance that is related to the substance’s microscopic structure. For example, the forged park gate in your city was produced thanks to the malleability and ductility of metals, which are related to the ability of shifting of layers of metal cations, while fire extinguishing powders use the high boiling point of compounds related to their regular ionic and covalent structures. This also applies to polymers. The purpose of this review is to summarise and present information on selected food-related biopolymers, with special attention on their respective structures, related properties, and resultant applications. Moreover, this paper also highlights how the treatment method used affects the structure, properties, and, hence, applications of some polysaccharides. Despite a strong focus on food-related biopolymers, this review is addressed to a broad community of both material engineers and food researchers.

## 1. Introduction

Various applications of polymer-based materials, ranging from films and coatings for food and packaging, through to the delivery of health-promoting bioactive compounds, to food–body interactions manipulated via nanotechnology, have recently gained interest in fields such as nutrition, medicine, and pharmacology. The number of previous and recent reports call for an urgent need to compile and review the multiple structural aspects of these important materials with relation to their properties and both current and future applications [1,2,3,4]. The present review undertakes this ambitious task, with a special focus on food-related uses, describing some of the most prevalent and safest biopolymers, with the widest spectra of applications. This way the reader is given the wide overview of biopolymers, such as starch, pullulan, carrageenan, pectin, agar, nucleic acids, collagen, and gelatine, collected in one place and providing an excellent starting point for further studies.

## 2. Starch

### 2.1. Structure

Starch is a branched homopolymer of glucose, usually synthesized in plant amyloplasts to store the energy gained from photosynthesis. Starch comprises two types of polymers: amylose and amylopectin [2]. Amylose (Figure 1) is a linear molecule of glucose units linked by α 1-4 glycosidic bonds, slightly branched by α 1-6 linkages, while amylopectin (Figure 2) is a highly branched polymer consisting of relatively short branches of α 1-4 glycopyranose that are interlinked by α 1-6 glycosidic linkages approximately every 22 glucose units [3]. The larger molecule of amylopectin is branched non-randomly and this organized branching nature of amylopectin creates symmetry and a type of uniformity responsible for the crystallinity of the starch [4], while amylose is found in the amorphous lamella of the starch granule. Observed under a microscope, starch shows birefringence under polarized light, and the specific “Maltese cross” indicating a radial orientation of the macromolecules [3].

### 2.2. Properties and Applications

Starch is one of the most abundant biopolymers. It is completely biodegradable, renewable, and inexpensive (e.g., much cheaper than polyethylene). Moreover, it is colourless, but easy to colour; odourless, but easy to flavour; edible though tasteless; and easy to mix with other components [5].

Starch is the major source of energy in plants (up to about 90% dry matter). It is usually accumulated in tubers, seeds, and roots. Most starches for the industry are obtained from maize, wheat, cassava, potato, and the recently rediscovered amaranth and quinoa [5]. Starch is the main energy storage carbohydrate in wheat and constitutes approximately 60–75% of grain and 70–80% of flour. It has a vital role to play in human nutrition as it contributes to 90% of caloric intake in developing countries and more than 50% in the Western world [6].

A starch’s nutritive value is linked with the ease of its digestion in the small intestine. Its dietary value is defined by the content of rapidly digestible starch (RDS), slowly digestible starch (SDS), as well as resistant starch (RS). RDS causes a rapid and sharp postprandial rise in blood glucose levels and is therefore not recommended for people with diabetes and obesity. SDS is the fraction of starch that slowly but completely hydrolyses in the small intestine. Due to its stabilising effect on glucose level, it is physiologically more preponderant than RDS. Some benefits of consuming SDS include the prevention and treatment of type II diabetes because it promotes feeling of satiety through the metabolic response, namely postprandial low blood glucose and insulin levels [7].

RS, in turn, is defined as starch and its breakdown products that are not absorbed in the small intestine of a healthy human. RS in the diet exerts a positive effect on the human body as it stimulates the growth of beneficial microbiota and lowers colonic pH, postprandial blood glucose, and cholesterol levels. RS is ideal for use in ready-to-eat breakfast cereals, snacks, pasta, noodles, baked goods, and fried foods, and allows for food to be labelled as simply “starch with added nutraceutical benefits”. Its content in food can be increased by modifying processing conditions such as pH, temperature, number of heating and cooling cycles, and freezing and drying times. RS improves the tenderness of some products, resulting in a better mouthfeel than products made with traditional fibres. RS exhibits physiological properties that may reduce the risk of several diseases, including colon cancer and diabetes, and may be very useful in controlling diabetes and obesity. Products fortified with RS are willingly accepted by consumers for the unique physicochemical properties of RS and its bland taste. RS is for example retrograded starch, produced as a result of cooking and cooling of starchy foods, as well as amylase–lipid complexes produced during thermal processes of starch in the presence of lipids. The potential prebiotic effect of RS, resulting from its ability to be fermented by the colon microbiota, leads to the production of short-chain fatty acids. Chemical, physical, and enzymatic methods to produce SDS and RS with specific structures provide opportunities to tailor both fractions for health benefits [7,8].

Starches are classified based on the proportions of amylose and amylopectin: starches with 25–30% amylose and 70–75% amylopectin are called “normal” starches; starches with very high levels of amylopectin (98–99%) are named “waxy” starches; a third group contains starches with high amylose content (50–70%) and has no specific name [8].

The ratio of amylose to amylopectin in starch profoundly impacts its physico-chemical properties or starch-based industrial products’ features, which in turn affect their functionality and potential applications. Higher amylose content implies increased film strength, while the presence of highly branched amylopectin leads to the formation of films with poor mechanical strength, which needs to be enhanced by adding sorbitol, glycerol, or other plasticizers [9].

Ultrasound assisted (UA) treatment increases swelling power of sweet potato and cereal starches. The damage of the starch granules helps water to penetrate them easily, increasing the starch solubility. The longer the time of UA treatment, the lower the starch peak and final viscosities. Disintegration of the starch granules can cause the reduced pasting viscosities; hydration and disassociation of starch chains are caused by water penetrating through the pits and pores on granule surface. Ultrasound greatly increases the RDS content of gelatinised maize starch. A decrease in crystallinity index and damage to starch granules allows for easier penetration of the enzymes and can be responsible for the increase in the enzymatic hydrolysis. UA-treated gelatinized corn starch dispersions produce starch films with better transparency and a more cohesive structure than untreated ones. UA treatment destroys non-solubilised part of gelatinised starch and releases more amylose to form stronger film [1].

The development of other non-thermal techniques such as pulsed electric field (PEF) and cold plasma (CP) treatment have attracted more attention recently. Various field intensities of PEF applied to starch resulted in its narrow molecular size distribution suggesting that chain breakage did not occur in this process. CP treatment in turn only partially decomposed starch granules and did not form starch nanoparticles, whereas they were formed when the combination of CP with UA was used [1].

Unmodified starches, also known as native starches (NS), are insoluble in water, show inadequate thermal dissociation, and suffer readily from retrogradation and syneresis. Furthermore, the paste and gel of NSs are unstable under stress, at high temperature, and high pH. There are many disadvantages associated with NSs: unmodified sweet potato products which contain NS are unacceptably hard and have poor transparency, whereas cereal NSs make cohesive, rubbery, weak-bodied pastes and form unwanted gels while boiled. However, as NS contains ubiquitous hydroxyl groups and simple glycosidic bonds, it can be easily modified physically, chemically, enzymatically, or biotechnologically, or using a combination of these methods. The aim of the modifications is to incorporate specific features enhancing cooking properties, gel clarity, sheen, texture, adhesion, and film formation, increasing freeze–thaw stability or reducing syneresis, retrogradation, and gelling tendencies. Starch amphiphilicity, hydrophobicity, mechanical strength, and thermal stability are among the properties that can be improved as a result of starch modification [4]. The comparison of structure, properties and applications of unmodified and UA-treated starch is presented in Table 1 below.

The emulsion capacity of NS is somewhat low because of relatively large granule size (>1 μm) and low surface hydrophobicity. Dry heating or chemical modification can increase granule hydrophobicity but heat modification does not require any particular labelling when used in food applications, what makes it more attractive. However, the application of starch granules as emulsifiers is limited by the smallest possible granule size. In turn, nanocrystals of starch (size range 40–100 nm), obtained by NS hydrolysis, are very efficient stabilisers of continuous water emulsions [3].

Starch is commonly utilized in numerous industries such as food, paper, textile, chemical, and pharmaceutical, where it plays various roles such as adhesive, stabiliser, thickener, as well as bulking, gelling, and water holding agent [6]. Starch in various forms can be blended with conventional polymers to produce other bio-based polymers [9].

Starch is used to fabric biodegradable films which are translucent or transparent, colourless, flavourless, and tasteless. Improving mechanical-handling properties and structural integrity of food as well as acting as barriers to flavour compounds, gases, and water vapour blocking (WVB) are amongst traditional tasks of conventional packaging and need to be performed by edible starch-based films and coatings. Starch-based coatings can be successfully used for fresh fruit and vegetable packaging, e.g., the edible composite coating based on tapioca-starch with κ-carrageenan coating was applied onto fresh pumpkin to extend the shelf life. Corn starch-based coating with sunflower oil applied to Brussels sprouts allowed to reduce loss of moisture, polyphenols, and vitamin C, and to extend retention of colour and firmness. The effectiveness of emulsion coatings based on corn starch, methylcellulose, and soybean oil was noted in slowing down the hydration process of bakery products such as crackers in high relative humidity conditions. Similarly, the application of emulsified coatings from pea starch, whey protein isolate, and carnauba wax to walnuts and pine nuts confirmed that edible packaging can prevent the oxidation and hydrolytic rancidity of nuts. In addition, the coated nuts received higher sensory scores than the control group [10]. There are examples of starch use for meat and poultry packaging, e.g., starch-based film (with chitosan, alginate, and pectin) was used on fresh turkey [11].

Another currently explored novel application of starch is in the immobilisation of enzymes, proteins, or drugs to improve catalytic activity of enzymes used in food technology or to create more efficient drug delivery systems. The incorporation of excess hydroxyl groups into the starch backbone is necessary for numerous functional groups, such as polyamines, ester, ether, and carboxyl groups to be embedded, enabling starch to bind conveniently to ligands such as amino acids, proteins, and enzymes. Some examples of immobilisation using starch are presented below. The urease was successfully immobilised on starch molecule, i.e., Dialdehyde Porous Starch, by chemical and physical methods. The L-asparaginase was solidly immobilised on the poly(2-hydroxyethyl methacrylate)-starch (pHEMA-S) matrix by physical method and gained different benefits such as higher activity and enhanced storage and thermal stability. Complementary research was reported on immobilising L-asparaginase on poly(methyl methacrylate)-starch composites (pMMA-S) instead of (pHEMA-S) prepared by emulsion polymerisation technique. According to these results non-starch pMMA had not as good thermal and hydrophilic properties as the starch composites. Design of medicine delivery and biocatalyst immobilisation systems would not be possible without nanocarriers benefitting the sectors of food industry pharmacy, medicine, bioreactors, and biosensors. The immobilisation of lipase on the magnetic nanomaterials of dialdehyde starch was also performed. Immobilised lipase had storage stability and recycling rate at 82.5% and 53.6% of the native enzyme, respectively. The efficiency of the biocatalyst increased with an external magnet used to separate and purify final reaction product [12].

## 3. Pullulan

### 3.1. Structure

Pullulan (Figure 3) is a homo-polysaccharide consisting of maltotriose residues linked by α 1-6 glycosidic bonds and produced by the fungus *Aureobasidium pullulans* using starch as a substrate [13,14].

### 3.2. Properties

Pullulan is highly soluble in water and displays unique film-forming properties. Films formed by pullulan being impermeable boundaries to gases and oils are perfect as edible coating materials. Furthermore, they are transparent, tasteless, odourless, and highly permeable to water [13]. Pullulan of food, pharmaceutical and cosmetic grades is listed on the United States Pharmacopoeia—National Formulary and Japanese Pharmacopoeia [15].

### 3.3. Applications

With its superior film-forming performance, good gas blocking characteristics, and desired mechanical properties, pullulan is widely used to manufacture biopolymer films [13,14].

While the application of pullulan brings numerous benefits, the high price has hindered its widespread use in industry. This disadvantage is overcome by mixing pullulan with other biopolymers, e.g., alginate, chitosan, cellulose derivatives, starch, and casein. As a result, pullulan films with improved physicochemical and mechanical properties were produced. From the above-mentioned polysaccharides, starch is cost-effective, renewable, and easily biodegradable [13]. Similarly, blending pullulan and carrageenan improves physical properties as they supplement each other’s shortcomings. This blend-film was further improved by the use of copper-sulfide nanoparticles and limonene as fillers (homogenously spread in the matrix film). This way, the mechanical characteristics of received film, namely tensile strength, elongation at break, and Young’s modulus, were profoundly improved, whilst the thermostability remained unchanged. The WVB was insignificantly enhanced, with hydrophobicity remaining unaltered. The films manufactured with the addition of various probiotic strains also exhibited high antimicrobial activity against pathogenic microorganisms, and thus they may be applied to monitor their growth in food [14].

Pullulan and its derivatives have numerous advantages, such as high solubility in water, homogenous dispersity, as well as non-toxicity and plasticity. Moreover, they are biodegradable, biocompatible, and thermally and chemically resistant. These features make pullulan ideal as a support for immobilisation of enzymes and anti-cancer drugs. In 1981 Hirohara et al. were granted the US Patent [16] for their novel approach to immobilisation of highly specific and exhibiting great catalytic activity enzymes by pullulan gel.

The immobilisation of *Burkholderia cepacia* lipase was performed on a chemically modified pullulan matrix that occurred to be a perfect carrier as the enlarged particle size consequently increased the adsorption surface for the lipase. As a result, the immobilised lipase shortened the reaction equilibrium time significantly. In another research, a green method of β-glucosidase covalent immobilisation was tested. Poly-aldehyde pullulan was applied as a cross-linking agent. Amino-tannic acid-modified Fe_3_O_4_ magnetic nanoparticles were used as a biocompatible nanoplatform. As a consequence, enzymatic activity of the enzyme was significantly improved [12].

## 4. Carrageenan

### 4.1. Structure

Carrageenans and agarans constitute nearly half of the algal dry matter and they are the major cell wall carbohydrates of marine red seaweeds of the *Rhodophyceae* class. As they are biocompatible, biodegradable, and low toxic their importance as components of pharmaceuticals and beauty products as well as in nutrition is growing [17].

The term carrageenan (CG) includes compounds from a class of heavy molecular weight linear sulfated carbohydrates containing α-D-galactopyranose and 3,6-anhydro-galactopyranose molecules linked together by α 1-3 and β 1-4 glycosidic bonds (Figure 4). CG is negatively charged and hydrophilic and its structure is similar to natural glycosaminoglycan.

Kappa-CG, ι-CG (iota), λ-CG (lambda), μ-CG (mu), ν-CG (nu), and θ-CG (theta) are six main kinds of carrageenans. Amongst them, κ-, ι- and λ- are the most important and commercially applicable types of CG. The amount and distribution of ester sulfate groups and the 3,6-anhydro-galactopyranose units in CGs affect primarily the properties of the compounds. Kappa-, ι- and λ-CG have 1, 2, and 3 negatively charged ester sulfate groups per dimer unit, respectively, but λ-CG has no 3,6-anhydro-D-galactopyranosyl linkage (Figure 5). Mu-, ν-, and θ-CG are biological derivatives of κ-, ι-, and λ-CG, respectively [14,18,19,20].

### 4.2. Properties and Applications

All CGs are soluble in water but do not dissolve in organic media. The number of sulphate groups, and the resulting Na+ and K+ cation equilibrium strongly affects CG water solubility, viscosity, and gelling ability. Clinical studies confirmed CGs being very low or non-toxic, and causing no developmental malformations [18]. As per low toxicity, the safety of CG as food additive was analysed in a large number of studies and concluded that a CG daily dose of up to 75 mg/kg body weight in the human diet was safe [21].

Even a small amount (1% by weight) of curcumin added to CG functional films improved the surface hydrophobicity, swelling ratio, as well as UV-blocking and WVB properties. Additionally, the CG composite films gained antioxidant activity and some antibacterial properties, and could be applied in the form of coatings for active food packaging [22].

Similar results were found when κ-CG edible films were made with orange essential oil and trehalose. Such films could also be used for active food packaging, as blocking property of UV and visual light was greatly improved by the addition of orange essential oil and trehalose. Moreover, the packaging film gained meaningful antimicrobial activity against *Staphyloccocus aureus,* wherein the higher the content of κ-CG in the film, the higher the antimicrobial activity against all studied microorganisms [23].

In search of new applications, the effect of CG in yoghurt formulation was tested. It was found that although CG (similarly to xanthan gum) made skim yoghurt formulation more solid and viscous, but at the same time syneresis occurred and positive organoleptic features became worse [24].

Kappa-CG can be used to produce the composite hydrogel with methacrylate. Addition of κ-CG increased gel water uptake over 3 times which makes such a hydrogel classified as superabsorbent. It also significantly improved mechanical properties of hydrogel which, as a result, gained features useful for biotechnology and drug formulation or sustained drug release [17,19].

Carrageenan is also used by some manufacturers as gelling agent to produce toothpaste hydrogel. Adding limonene as filler improves mechanical properties of such a gel (this is the authors’ observation based on the local Polish dental hygiene product market).

The CGs antioxidant property was screened by scavenging assays of such reactive oxygen species (ROS) as 2,2-diphenyl-1-picrylhydrazyl radical as well as superoxide and hydroxyl radicals. Similarly, the CG capability of ROS cleaning could also be confirmed by the increase in superoxide dismutase, catalase, glutathione peroxidase, or Fe^2+^ chelating ability. For example, ι-CG from *Solieria filiformis* manifests antioxidant activity by preventing glutathione consumption and reducing levels of malondialdehyde and haemoglobin, while κ-CG shows outstanding capacities of free radicals scavenging (including all listed above) and Fe^2+^ chelating activity in vitro [17].

Sulfate groups in CGs can neutralize positive charges on the surface of host cells, thereby impeding viral uptake, penetration, and detachment, wherein the higher the number of sulphate residues and degree of polymerisation, the stronger the antiviral property. Various types of CG have various antiviral inhibitory activities against various viruses. For example, fully oxidized κ-CG was proven to have worse antiviral property against HSV-1 and HSV-2 (herpes simplex virus) than its partially oxidized counterpart. In addition, the combination of CG and griffithsin (a non-antiretroviral HIV entry inhibitor) indicated strong possibility of interfering with HIV-1, HSV-2, and HPV, which would significantly reduce the risk of sexually transmitted infections in women. This result indicates a potential of enhancing of CGs antiviral activities through drug combination therapy. It was observed that CGs helped to block the connection of virus to receptors of host cells as a result of CG’s affinities for glycoproteins of viruses [17,20].

Historical records dated as early as the 1830s mentioned that “carrageen” or “Irish moss” (known as not specified mixture of coexisting naturally marine red seaweeds, *Chondrus crispus* and *Mastocarpus stellatus*) was often applied as medicine. Even in modern Ireland it is still thought to have antiviral properties and to be efficient in dislodging mucus, and therefore it is used in traditional curative infusions against common bronchitis, infections, and chronic coughs. High iodine content and traditional presence of seaweed in the diet was believed to be related to initial low level of infection caused by newly emerging SARS-CoV-2 in Japan, specifically in Hokkaido. Admittedly, the number of these infections increased dramatically afterwards; however, it seems that seaweed helped Hokkaido population to reduce contagion in an initial period, though soon such additional actions as isolation or social distancing were required. Both children and adults with virus-originated cold, treated with a CG-based nasal spray had the disease duration shortened with faster removal of virus. It is worth mentioning that there is a commercially available ι-CG-based nasal spray named Bisolviral^®^ sold by Sanofi Aventis [25].

In the past many reports confirmed both κ-CG and ι-CG activities against influenza A virus in vitro as well as positive therapeutic effects in vivo, while the antiviral potential of λ-CG was seldom examined. Therefore, the special attention was paid to check if λ-CG is able to protect against both influenza A and B viruses and/or SARS-CoV-2, as this compound has a higher sulfate groups number and is easier soluble in cold water than the other two CGs. An experimental study was run to prove whether λ-CG is active against various respiratory viruses, namely: influenza A and B viruses and SARS-CoV-2. The data revealed that not only did λ-CG suppress expression of viral proteins in invaded cells, but also inhibited the production of viral progeny in a quantity-proportional way. Taken together, these results suggested that λ-CG had significant antiviral potential against influenza A and B viruses in vitro. In vivo antiviral activity was assessed by observing body mass and lethality of infected mice for 15 days. The infected and untreated mice suffered from significant body mass loss and died not later than on 7th day, yet λ-CG administered as nasal spray at 5 mg/kg alleviated infection-induced body mass loss, resulting in a 60% survival rate. Firefly luciferase test using lysates of infected cells showed that λ-CG was capable of limiting infections of both SARS-CoV-2 and pseudoviruses derived from influenza A viral glycoproteins in a quantity-proportional way. Moreover, further examinations showed λ-CG reduced both the level of viral protein in cell lysate and the level of viral RNA in the supernatants. The above analyses unequivocally demonstrated the antiviral activity of λ-CG against SARS-CoV-2 [20].

Among numerous methods of chemical modification applicable to CGs carboxy-methylation seems to be the most commonly used, because it makes CG more soluble in water, what together with CG natural features of gel-forming and viscosity make it a very attractive compound for drug delivery and specifically for controlled drug release. Nevertheless, oral administration of CGs still risks unexpected immune reaction, harmful gastrointestinal (GI) effects, and other potentially unwanted organism’s responses [17]. The comparison of structure, properties and applications of unmodified and modified carrageenans is presented in Table 2 below.

## 5. Pectins

### 5.1. Structure

Pectin constitutes a very complicated and differentiated structurally group of anionic polymers. The chemical structure of pectin varies from plant to plant, tissue to tissue, and even within the cell wall [26], except that all pectins contain no less than 65% D-galacturonic acid (GalA) as repeating unit (Figure 6), bonded at the O-1 and O-4 positions.

Three major pectin polysaccharides, namely homogalacturonan (HG) and rhamnogalacturonan I and II (RG-I and RG-II), are well-described. Among them, HG is the most common one, with a linear homopolymer of GalA linked by α 1-4. RG-I makes 20–35% of pectin and has the backbone of the GalA disaccharide repeating unit bonded to rhamnose. RG-II forms around 10% of pectin and contains a skeleton of HG, with complicated side chains of different sugars such as apiofuranose, arabinofuranose, fucose, galactose, rhamnose, and more, attached to the GalA residues.

It is generally accepted that all pectin polysaccharides are covalently cross-linked, as harsh chemical treatment or digestive enzymes are needed when attempting to isolate them from each other. Admittedly, the specific linkages amongst them have not been well understood yet, but they are created as a result of pectin biosynthesis in plants [27,28].

HG contains α 1-4 linked GalA units that are partially methyl-esterified. The degree of esterification (DE) of GalA residues is applied to categorise pectins. DE is defined as the molar quotient of methyl-esters quantity and total number of GalA units, expressed as a percentage. It strongly affects formation of emulsion by pectins, their gelling capability, and surface tension. High methoxy (HM) and low methoxy (LM) are two categories of pectin resulting from DE: HM pectins require both a high percentage of another dissolved substance (55–75%) and acidic environment (pH 2.50–3.50) to gel, while LM pectins need neither [26].

Most commercial pectins, typically HM [26], are side-products of fruit juice industry, mainly made of citrus zest and apple residues, which contain about 20–30% and 10–15% pectin in their dry mass, respectively [29].

De-esterification is used to commercially obtain LM pectins from HM, despite LM being naturally present in fruit and vegetables. This process may be catalysed by acids (such as HCl), alkalines, or enzymes. However, wasting of many precious components, high process temperature, and ecological concerns are disadvantages of using acids. Bases de-esterification in turn is conducted at relatively low temperatures using alcoholic solution of ammonia. As a result, amidated low methoxy (ALM) pectin is obtained. GalA units, de-esterified enzymatically (e.g., with pectin methyl-esterase), are distributed randomly, which is important advantage of this method [1,26].

### 5.2. Properties and Applications

According to World Health Organisation (WHO) and Food and Agriculture Organisation (FAO) pectin is an ecological, healthy, natural, and nutritional food ingredient [29].

In human organism pectins can lower serum cholesterol and serum glucose levels. Pectins have also anti-cancer effects, particularly on the colon cancer, because of their prebiotic potential meaning fermentation of pectins into short-chain fatty acids by gut microflora [28], such as *Eubacterium*, *Bacteroides*, *Bifidobacterium*, *Clostridium*, *Erwinia*, and *Escherichia* strains and thus stimulating of the immune response. Unless fermented, pectins would survive the digestive system nearly intact [26].

The capacity of pectin gels to swell at low pH is an important benefit in obesity therapy. Once in the aqueous environment of gastric fluids, pectin gels swell, thereby stuffing the stomach long before digestion, leading to an extended feeling of satiety [27].

Gelation is one of the main functions of pectin, that is itself soluble in water, but when converted into a hydrogel, it becomes an elastic three-dimensional (3D) network of polymer chains which swell but do not dissolve in water [27]. Different gelling performance characterize HM and LM pectins; however, the same macroscopic properties, such as polymer composition, moiety size, and conformation, influence the resultant gel features. HM pectin makes gels at low pH and a high proportion of sucrose or analogous co-solvent. High sucrose proportion limits water activity, thus favouring chain-to-chain over chain-to-solvent interactions, whereas low pH promotes the carboxyl groups protonation, hence minimizing electrostatic repulsion. Stabilisation of HM pectin gels is predominantly achieved by electrostatic hydrogen bonding and methyl esters’ hydrophobic interactions, while in case of LM–by ionic bridges of Ca^2+^ between GalA carboxyl residues of adjacent molecules [26].

Alongside many advantages of pectins, they lack reproducible performance due to the large diversity of molecular structures, what complicates quality assurance in production. This problem can be minimised by: introduction of new techniques of pectin extraction and purification as well as chemical and physical modifications of pectins. Chemical modifications of pectins involve applying substituents, such as alkyl and thiol groups [26]. One of the physical modification methods is ultrasound treatment, which reduces values of pectin gel textural properties (hardness, gumminess, cohesiveness, resilience, and chewiness) with its time as pectin molar mass decreases. Simultaneously, UA depolymerised pectins exhibit higher antioxidant activity, because new antioxidative groups are created as a consequence of this modification. Moreover, the reducing radicals are formed during water sonolysis [1]. Additionally, UA modified pectin gels have better optical properties, while UA treated pectin exhibits improved performance as emulsifier in an oil-in-water emulsion systems [29]. Another method of physical modification applicable to degrade pectins is high pressure processing, leading to the reduction of pectin viscosity, average molar mass, and particle size [1].

Pectin is widely applied in the food technology to produce bakery fillings, confectionary products, fruit juice, jams and jellies, and to apply films and coatings on fresh fruit or vegetables Traditional functions of pectin include: thickening and gelling, stabilising, texturizing, and emulsifying [27,28].

Pectin is often used as stabiliser in acidified milk drinks, with pH, ionic strength, charge density and pectin concentration being the most important factors affecting its interactions with casein. Pectin adsorbs to the surface of casein molecules through electrostatic interactions between their carboxyl groups and the cationic amino acid residues of casein at or below pH 5. Fibre added to pectins further stabilizes the casein gel network as a filler. Emulsifying and stabilising properties of the resulting from Maillard reactions pectin combinations with proteins are excellent. Pectin–protein ability to create emulsion is controlled by the protein part, feruloyl, and acetyl groups, while the pectin HG skeleton and RG-I side chains play a major role in emulsion stabilizing. The further advantage of these systems is that proteins exhibit significantly higher ability to anchor at the oil–water interface than pectins, although protein-stabilised emulsions have worse acid stability [28].

## 6. Agar

### 6.1. Structure

Agar (or agar-agar) together with carrageenans (described above in detail) are important cell wall polysaccharides obtained from marine red seaweeds of the *Rhodophyceae* (red algae) class [18].

Agar is a mixture of polysaccharides: linear agarose—constituting of about 70% of agar—and acidic agaropectin. The agarose monomeric repeating units are neoagarobiose and agarobiose consisting of β-D-galactopiranose and 3,6-anhydro-α-L-galactopiranose bonded by β 1-4 and α 1-3 glycosidic linkages, respectively (Figure 7).

The agaropectin repeating units are the same, but about 8% of the C-2 or C-6 positions of monomers are substituted by sulphonate, methoxy, pyruvate, or glucuronate residues, which are a part of the agar skeleton that make it useful in protein immobilisation and electrophoresis [12,30].

### 6.2. Properties and Applications

Agar is a non-toxic and inexpensive polysaccharide. It exhibits a strong ability to gel, does not react with proteins and acids, and is stable in basic solutions, which make agar useful in biotechnology. Moreover, even at very low concentrations of 0.1%, agar gels are stable at temperatures up to 85 °C [18]. Even more interestingly, proteolytic enzymes present in some fruits do not decompose agar making this polymer a very efficient equivalent of gelatine (that would be decomposed by those enzymes) in production of fruit jams.

Agar and potato starch were used to develop colorimetric pH indicator films, in which they were carriers to immobilise natural pigments, such as anthocyanins. Such an indicator can have a potential use as a meat spoilage sensor [31]. In order to estimate fish freshness an agar-based indicator film with the addition of a natural colourant derived from the root of *Arnebia Euchroma* (AEREs) was developed. The addition of AEREs caused some changes of the film properties such as increase in elasticity modulus, and tensile strength as well as reduction in water solubility, swelling coefficient, and WVB. The tests showed that the colour response of the label was consistent with the spoilage threshold of total volatile basic nitrogen and total viable count—both periodically determined for the fish sample. Such agar-based films can be a comfortable, undamaging, and viewable way to monitor spoilage of meat or fish during their shelf life [32].

As in the case of CG films described above, even a small amount (1% by weight) of curcumin added to agar-based functional films improved the surface hydrophobicity, swelling ratio, as well as UV-blocking and WVB properties. Additionally, the agar composite film gained antioxidant activity and some antibacterial property, and could be applied for active food packaging in the form of coatings [22].

Agar concentration has an effect on the properties of polysaccharide-based films plasticised with deep eutectic solvents (DES). Agar films are produced with successful application of choline-based eutectic mixtures (with urea or glycerol). An analysis of the films’ mechanical properties and sorption isotherms indicates that the higher the concentration of agar in the film, the higher its tensile strength and WVB. In turn, agar nanofibres are produced by electrospinning technique using choline chloride with urea at 1:2 molar ratio as an alternative to water solvent and poly(vinyl alcohol) as co-blending polymer. Such fibres are characterised by higher spinnability and viscoelasticity compared to fibres produced traditionally in water solution. Historically, the mixture of choline chloride with urea was first presented in 2003 as an original concept of DES [30].

Agar is used in electrophoresis applied to the separation of macromolecules, such as nucleic acids or polypeptides, using the principle of electrophoretic mobility depending on molecular mass, size and electrostatic charge. Nucleic acid fragments can be separated in an agarose gel matrix by chain length (mass and size), while the separation of proteins is performed by charge or molecular size. An agarose gel is a 3D structure formed from agarose helices coiled in bundles and held together by hydrogen bonds, with channels and pores, which biomolecules can move through [33].

Agar is hydrophilic, lyophilic, inert, and produces solid but reversible gels, what makes it attractive material to encapsulate enzymes. Encapsulation allows to preserve conformation and activity of native enzymes. For example, agar turned out to be better than polyacrylamide and gelatine in terms of the efficiency of the Trametes Versicolor IBL-04 laccase immobilisation performed on the agar gel using encapsulation. Optimisation of agar concentration, temperature, pH, and other immobilisation conditions allowed for 79.7% biocatalytic activity to be maintained at 55 °C, pH 7.0, and agar proportion of 3.0%. Laccase immobilised via encapsulation is used in degradation of toxins and recalcitrant micro-pollutants, pulp bleaching, clarification of fruit juices, and valorisation of lignins. Besides encapsulation there are other numerous enzyme immobilisation techniques on the agar matrix and agarose beads, namely entrapment, covalent, adsorption, and cross-linking techniques. Similar to encapsulation, the entrapment method advantage is inducing no structural alterations that could affect biocatalytic activity of entrapped enzyme. This technique is used in an agar bead column reactor introduced to immobilise polygalacturonase applicable in fruit and vegetable processing and obtained from *Streptomyces halstedii*. In turn, covalent bonding method is used for immobilising β-D-galactosidase on novel grafted agar discs [12].

## 7. Nucleic Acids

### 7.1. Structure

Other biopolymers present in diet are nucleic acids, deoxyribonucleic acid (DNA), and ribonucleic acid (RNA).

The DNA monomers called nucleotides are composed of a 5-carbon sugar–deoxyribose, a nitrogenous base (either the double-ring purine: adenine or guanine, or the single-ring pyrimidine: cytosine or thymine) and one or more phosphate groups that are acidic (Figure 8). Nucleotides are bonded via a phosphodiester linkage between the 3′-hydroxyl group of one deoxyribose and phosphate group bonded to 5′-hydroxyl of the next deoxyribose. DNA is composed of two polynucleotide strands running in opposite directions and forming a right-handed α-helical secondary structure due to hydrogen bonds being formed between pairs of complementary nitrogenous bases (adenine and thymine; guanine and cytosine). DNA is wrapped around the histone proteins forming the nucleosome. These are then folded upon themselves to form a fibre, which is then folded again eventually forming the chromatid [34].

RNA is structurally similar to DNA, being a chain of similar monomers. RNA’s nucleotides are composed of a 5-carbon sugar–ribose, a nitrogenous base (either the double-ring purine: adenine or guanine, or the single-ring pyrimidine: cytosine or uracil), and a phosphate (Figure 9). RNA is less stable than DNA and forms secondary structures in some rare cases [34].

### 7.2. Properties and Applications

These long-chain polymers of nucleotides are natural components of food since food is derived from once-living organisms [35]. Daily intake of foreign DNA is estimated at around 0.1 g to 1.0 g [36].

Digestion of nucleic acids begins in the stomach. Pepsin enables DNA separation from histones. Additionally, it was shown that it may also digest nucleic acids [37]. The digestion of nucleic acids is then continued in the intestine where they are metabolised by endonucleases, phosphodiesterases, and nucleoside phosphorylase into oligonucleotides, nucleotides, and free bases [37,38]. Some of the products formed can be absorbed by intestinal endothelial cells and used to synthesize nucleic acids in the body [37]. Based on the daily DNA intake given earlier it has been estimated that around 0.5 μg DNA (which corresponds to 10^10^ DNA fragments of 1000 base pairs) is absorbed each day using the human intestinal epithelial cell line CaCo2 model [39]. This transport of foreign DNA via the intestinal barrier has a very important role. Immunosensory cells lining up the GI tract can detect the bacterial unmethylated CpG motifs which triggers defence mechanisms [36].

Nucleic acids are sources of purines, which on degradation produce uric acid [40]. Foods with purine content ≥ 1000 mg/kg are defined as purine-rich foods. These include seafood (fish and shellfish), red meat, poultry, legumes, some vegetables (cauliflower, spinach, and asparagus), and fungi (shiitake mushroom and hazelnut mushroom) [41]. Studies have shown an increased serum urate concentration caused by a purine-rich diet, posing a risk of gout [42], especially in the case of animal-derived food (red meat and poultry) [41]. On the other hand, exogenously supplied nucleotides may be beneficial in conditions characterized by a greater demand for synthesis of nucleic acids, such as periods of decreased protein intake, rapid growth, immunosuppression, or gut injury [43]. Moreover, it has been shown that dietary nucleotide supplementation (nucleotides and RNA) improves some symptoms associated with irritable bowel disease, such as abdominal pain [44].

Cooking, baking, roasting, and frying all lead to DNA degradation. Elevated temperatures result in depurination, deamination, strand scission, and irreversible loss of DNA secondary structures. Additionally, DNA oxidation and hydroxylation may occur. Therefore, processed food is known to contain low content of DNA, because DNA molecules are destroyed and fragmented as a result of chemical processing at low pH, high pressure, and temperature [36].

Available studies present sufficient evidence for the presence of small diet-derived DNA fragments in the blood of humans [45,46]. However, it is very unlikely that these fragments are large enough to carry complete genes. Moreover, integration of these fragments in the genome of intestinal epithelial or other cell types is unlikely and mechanisms responsible for the foreign DNA transcriptional silencing exists such as inactivation by methylation of cytosine residues at CpG dinucleotides [36]. Interestingly, derived from plant-food noncoding microRNAs have been shown to survive the digestion process and enter the body of the consumer potentially impacting the gene expression processes [47,48].

With time more and more products derived from genetically modified (GM) crops have become a part of the human diet and livestock feed raising concerns over their safety. However, according to the literature, GM food-derived DNA is not characterised by a greater potential for being uptaken and integrated as compared to the genetically unmodified food-derived DNA. Experiments under optimised laboratory conditions have shown that GM plant-derived DNA fragments can be assimilated by single bacterial species [49], posing a risk of transgenes transfer to bacteria inhabiting the GI tract. However, there are not enough in vivo studies available yet to completely prove GM crop-derived DNA integration into the gut microbiota of mammalian species [36].

## 8. Proteins

### 8.1. Structure

Proteins are the building blocks of life and therefore are an essential part of healthy diet [50]. The basic units of all proteins are 20 amino acids, mainly L-amino acids. The amino and carboxylic acid groups of the amino acids are bonded to the same carbon atom referred to as the α-carbon atom. All proteins are chiral macromolecules, showing optical activity due to the presence of chiral atom(s). Amino acids join together from the N-terminus to the C-terminus via the peptide bond to produce the polypeptide. The primary structure of proteins is defined as the order of amino acids in the polypeptide chain. The secondary structure of proteins results from folding of the polypeptide chain. This can take the form of the α-helix (a single right-handed helix-shaped chain stabilised by hydrogen bonding between peptide groups) or β-sheet (at least two chains in parallel or anti-parallel fashion with hydrogen bond between peptide groups of two different chains). The tertiary structure of proteins is the arrangement of the secondary elements in three dimensions due to hydrophobic and ionic interactions, as well as hydrogen and disulphide bonds. In the case of proteins composed of more than one polypeptide chain the relative arrangement of these chains is referred to as quaternary structure [51]. The type, number, order, orientation, and side chains of the amino acids as well as interactions among them affect the physicochemical properties of proteins, such as surface hydrophobicity, net charge, and the reactivity of functional groups [52]. Proteins have an essential layer of ordered water molecules on their surface. They are not very stable polymers as they are easily denatured by extreme pH, rise in temperature, or organic solvents, which remove the essential water from them [51].

### 8.2. Applications

There are two main types of dietary protein sources: animals and plants [53]. Plant proteins may have insufficient amount of some essential amino acids, which cannot be synthesized in the human organism and therefore should be provided in the diet. Moreover, plant-derived proteins are usually less digestible and less bioavailable than animal proteins [54]. On the other hand, the production of conventional animal-based proteins to meet the rapidly growing global demand is challenging. Therefore, there is an increasing need for alternative, more sustainable protein sources, such as edible insects [53], microalgae, seaweeds [50], or even synthetic food biology [55].

When it comes to food, proteins have the ability to form gels and films, stabilise emulsions and foams [56]. The functional properties of proteins (gelation, solubility, thermal stability, emulsification, and foamability) depend on the molecular size, shape of proteins, physicochemical properties, as well as processing conditions [52]. Globular proteins, which are water-soluble, approximately spherical and the most abundant proteins in nature [57] and in foods, unfold on heating, at extreme pH, and ionic strength. These denatured chains, when present at sufficiently enough concentrations, aggregate forming thermally irreversible gels. As a result of the interactions between the hydrophobic domains of the unfolded chains, uniform, finely stranded 3D network structures are formed [56]. Fibrous proteins, as water-insoluble long fibres or sheets of parallel polypeptide chains [57], are able to form thermally reversible gels. One example is gelatine produced by acid/alkali treatment of collagenous materials. At high temperatures, gelatine molecules are disordered in solution but on cooling to around room temperature the molecules partially reform the triple helical structure of collagen in the process of thermally reversible coil–helix transformation [56].

Some proteins are amphiphilic and consequently they are able to adsorb at the interfaces of foams and emulsions, stabilising these mixtures. Protein molecules adsorb onto the newly created oil or gas droplets preventing their aggregation and coalescence. At pH other than their isoelectric point, proteins are net charged and thus they repel each other. At pH values close to the isoelectric point, stabilisation is brought about by enthalpic and entropic interactions between the adsorbed proteins resulting in steric repulsive forces. However, in most cases these interactions are too weak and that is why many studies focus on the combination of proteins and polysaccharides to enhance the stabilising properties. Moreover, several factors affect the stabilising function of proteins, such as their molecular size, conformation and solvent quality (e.g., pH which affects the proteins net electrostatic charges and conformation) [56].

Proteins application in food packages, including both edible and non-edible coatings and films was extensively reviewed recently [58,59]. A lot of scientific attention is devoted to enhancing properties of protein-based films and coatings being an attractive replacement for traditional petroleum-based films. Their superiority over those conventional films comes from their biodegradability, biocompatibility and, in some cases, edibility. The effects of protein type [60] as well as novel technologies used for film development [61] on the physicochemical, mechanical, and antioxidant properties of the resulting films are described in the literature. These properties can be also significantly improved using additives (such as plasticisers, nanoparticles of metals, metal oxides, or clay [62]) as well as when mixtures of different biopolymers are used, e.g., gelatine, whey protein, and chitosan composites [63]. A novel direction is the incorporation of active compounds such as phytochemicals further improving properties of protein-based coatings, e.g., increasing their antioxidant properties [64].

One of the most promising edible polymers to be potentially used as a food packaging material is whey protein in the form of whey protein isolate and concentrate. The whey film is characterised by a 3D gel-type structure being dry, colourless, odourless, flexible, and transparent with excellent mechanical and barrier properties [65]. The bovine milk-derived immunoglobulin-enriched whey film showed enhanced adhesion and tensile strength as well as better transparency [66].

Due to the environmental and ethical issues as well as rapidly growing world population there has been an increasing interest in plant-based protein composites which can replace traditional animal-based food. These meat analogues are plant-based products that resemble the animal meat in terms of the appearance, flavour, and the fibrous texture. The most challenging problem related to these substitutes is the texture and that is why much scientific attention is paid to the selection of the appropriate plant-based protein source as well as to processing approaches, such as thermal extrusion, freeze structuring, or fibre spinning [67]. These disruptive processes transform native globular plant proteins into filamentous aggregates or interactive fibres. Next, thickening, water-binding, and texture-enhancing agents are added to improve the texture and “juiciness” of plant protein-based meat analogues [68]. Plant-based protein sources commonly used in meat substitutes include soy, pea, wheat, and fungi proteins. They are also used in combinations. This may improve the nutritional and textural characteristics of the resulting meat analogues [67]. Unfortunately, some of the nutrients naturally present or added to plant protein-based meat substitutes are lost during the processing (blending, homogenization, and high temperature cooking). Moreover, meat alternatives usually contain more salt than their animal equivalents and some pre-processing treatment is needed to inactivate anti-nutritional components found in soy-based formulations. With these and many more limitations associated with plant protein-based meat analogues it is clear that further technical innovations and investigations should be performed [68].

Proteins are susceptible to oxidation which typically leads to the modification of the side chains of amino acids, with polymerisation and/or fragmentation being less frequent consequences. Hydrogen is easily lost from amino acids with side chains containing sulphur (cysteine and methionine) and/or aromatic rings (tryptophan, tyrosine, and phenylalanine). Food processing increase the oxidative stress due to the introduction of oxygen (e.g., grinding and mixing), removal and destruction of natural antioxidants (e.g., heat inactivation of antioxidant enzymes) as well as an increase in pro-oxidative factors (e.g., light exposure and thermal treatment). Conscious consumers expect “all natural” products and this increases the demand for new antioxidant technologies. The ability of proteins to inhibit lipid oxidation by being preferentially oxidised can be used to increase the oxidative stability of food (e.g., proteins with altered structure and consequently enhanced activity, antioxidant proteins introduced by genetic engineering, antioxidant proteins, and peptides as food additives) [69].

Apart from antioxidant properties, peptides and protein-rich fractions from hydrolysed proteins have many more biological activities as antihypertensive [70], antimicrobial [71], anti-inflammatory, or immune system modulating. The research interest in bioactive peptides is rapidly growing. Most works focus on peptides obtained by hydrolysis of whole food, protein concentrates, or isolates [72]. Hydrolysed dietary proteins or purified peptides used in food industry are mainly produced by enzymatic hydrolysis using microbial, plant, and animal proteolytic enzymes. Depending on the enzyme used, time, temperature, pH, and the source of proteins resulting peptides have different sequences and in consequence different biological properties [69]. However, to perform their biological functions these bioactive peptides must resist the digestion processes in the alimentary tract. In vivo data on their activity are very rare and dosage, absorption, pharmacokinetics, and side effects data are lacking. Moreover, evidence of the interactions of bioactive peptides with food matrices is limited [72].

Proteins from cereals do not have very high nutritional quality due to the low content of essential amino acids and limited water solubility. However, their nutritional value can be improved by hydrolysing them and releasing bioactive proteins. Many studies describe the antioxidant activity of cereal protein hydrolysates and peptides (wheat germ, rice dreg, rice endosperm, rice bran, corn gluten, hordein, barley gluten, oat bran, and wheat gluten proteins) in both in vitro and in vivo models (animal and food models, cell cultures, and chemical-based assays). Antioxidant activity can be altered by chemical modifications, e.g., conjugation with glucosamine and deamidation with citric acid [70].

### 8.3. Enzymes

Some proteins have ability to convert specific substrates into products at greater speed. They are referred to as biological catalysts—enzymes [73]. Enzymes are widely used in food industry to improve the taste, texture, rheology, and nutritional value of food products. Some examples of enzymes and their most up-to-date food-related applications are presented in Table 3. Many enzymes may be successfully used to process waste animal proteins resulting from food industry, which has been recently reviewed in [74].

Below, the reader will find further examples of proteins used extensively in the food industry with their structure–application relationship explored in details.

### 8.4. Collagen

#### 8.4.1. Structure

Collagen is the most dominant structural protein in vertebrates and invertebrates, comprising 30% of an animal’s total body proteins. Being the main component of the connective tissue extracellular matrix, it provides tissue integrity, mechanical strength, and flexibility [84,85].

On a molecular level, collagen exists as a right-handed triple helix, referred to as tropocollagen, formed by three parallel left-handed, PolyProline-II (PPII)-like polypeptide helices wound around one another and staggered by one residue (Figure 10). Both ends of the polypeptide are capped by short non-helical telopeptides or amino-terminal collagen cross-links. Usually, two of the polypeptide α-chains are chemically identical (α_1_) with the third chain being different (α_2_); however, collagens can also be homotrimers with three identical α-chains or heterotrimers with three different chains. The α-chains are around 1000 residues long with a molecular weight of around 100 kD. Basic and acidic amino acids are present at equimolar amounts [84,86,87,88].

The tight packing of the PPII-chains within the collagen triple helix gives rise to one of its signature characteristics; glycine (Gly) repeated at almost every third residue (meaning it accounts for around 30%) in a ${\left(\mathrm{Gly}-X-Y\right)}_{n}$ sequence. The second most abundant amino acid in collagen after glycine is Proline (Pro) accounting for 12% of all residues. The unusual imino acids hydroxoproline (Hyp) and hydrolisine are also present at high concentrations. The most common collagen amino/imino acid triplets are $\left(\mathrm{Gly}-\mathrm{Pro}-Y\right)$ or $\left(\mathrm{Gly}-X-\mathrm{Hyp}\right)$ triplets, with X and Y representing various other residues.

Glycine plays an important role in maintaining the spatial structure of the collagen triple helix as it facilitates the formation of hydrogen bonds between the α-chains in the case of a ${\left(\mathrm{Gly}-X-Y\right)}_{n}$ sequence, the amide group of Gly forms a hydrogen bond with the carbonyl group of X in the adjacent α-chains. One such bond is formed for every triplet [84,86,87,89]. Rather than every backbone carbonyl and amide group forming NH···C=O hydrogen bonds, further stabilisation of the triple helix is facilitated by a water network. This network forms hydrogen bonds between the still available carbonyl groups of Gly and Y situated on the same chain as well as between these carbonyl groups and the hydroxyl groups pointing out of the helix, e.g., Hyp on the adjacent chain. Water molecules might thus link residues on the same chain or on two different chains [90,91].

On the macroscopic level, tropocollagen molecules often self-associate into higher order structures such as staggered arrays of fibrils which associate into fibres, or hexagonal lattices, or networks (for further examples see [92,93]).

To date, 29 collagen types have been identified numbered with roman numerals in the order of discovery. Broadly, they can be divided into fibrillar and non-fibrillar collagens, the fibrillar collagens being most abundant (Types I, II, III, V, XI). It is worth noting that the ${\left(\mathrm{Gly}-X-Y\right)}_{n}$ sequence is only perfect in fibrillar collagens [86,89,94,95].

Commercial collagen and collagen-based products have traditionally been isolated from processing by-products of land-based animals such as cows and pigs and more recently from fish scales, skins, bones, and swim bladders [85].

#### 8.4.2. Properties and Applications

Collagen and its derivatives such as gelatine or collagen peptides have found numerous uses in the food, biomedical, beauty products, and nutraceutical industries [96,97,98]. This section focuses on food applications and is limited to the enhancement of food products, omitting the very broad area of nutraceuticals and their health benefits. To gain insight into the benefits of collagen supplementation it is advised to consult systematic reviews and meta-analyses such as [99,100].

Collagen is often used as a food additive, e.g., to improve the quality of meat products and to extend their shelf life. Schilling et al. [101] added pork collagen to boneless cured ham produced from pale, soft, and exudative (PSE) pork (containing less fat than conventional pork). Cooking loss decreased in non-PSE ham suggesting that collagen interacted with its myofibrillar structure binding water. Furthermore, the collagen reduced the expressible moisture, increasing food safety, in both PSE and non-PSE pork. This could be explained by the fact that collagen has a high swelling and water-binding capacity. Its chains form hydrogels due to covalent cross-linking and the formation of matrices that swell to an equilibrium volume in aqueous solution while preserving their shape [101]. Regarding cooking loss, the matrix’ swelling might block the exit of moisture and fat [102]. Shilling also demonstrated that collagen was able to increase the bind strength of 100% PSE ham [101]. Pereira et al. [102] obtained similar results investigating the effect of adding mechanically deboned poultry meat (MDPM) and collagen to frankfurter-type sausages by reducing the negative effect of MDPM on cooking yields and additionally demonstrating collagen’s capacity to improve colour and increase the desirable textural attribute of firmness. Chemical water retention and protein matrix swelling contributed to the cohesiveness of the batter and, consequently, its firmness.

Collagen gel extracted from chicken feet may be used to replace fat and prevent the negative effects of lipid oxidation when producing chicken sausages with lower fat content as shown in [103]. Rather than being a food additive itself, collagen can also be used as a carrier of food additives. One such example was described in 2007 by Waszkowiak and Dolata [104] who impregnated a collagen fibre preparation with rosemary extract to more evenly distribute it in wiener-type and liver sausages enhancing its antioxidative effects.

In the dairy industry, the gelation properties of collagen allowed Rama et al. to use it in its hydrolysed form of low molecular weight, referred to as collagen hydrolysate, to encapsulate probiotic lactic acid bacteria improving their survival rate when exposed to gastric juice containing pepsin, HCl and NaCl [105]. Continuing discussion on probiotic foods, adding collagen hydrolysate and açaí pulp improved the viscosity of milk beverages containing cheese whey, according to Rigoto et al. [106]. Znamirowska and colleagues showed that collagen protein hydrolysate is able to increase gel hardness and reduce syneresis, as well as improve *Bifidobacterium* Bb-12 survival during storage [107].

Collagen hydrolysate has a relatively low molecular weight and shows amphipathic properties due to being composed of both hydrophobic and hydrophilic amino acids, which means that it acts as a natural flocculant [108]. For example, collagen hydrolysates prepared from pig skin shavings have been used to clarify Chrysanthemum beverage (representing plant beverages) increasing transmittance from 55.4% to 96.2% [109].

Frozen bread technology also utilises collagen hydrolysate, but as ice-binding proteins (IBPs) or anti-freeze proteins (AFPs). These impede ice crystal formation by binding to the ice surface and then lowering the freezing point without affecting the melting point. Frozen bread goes through several freeze–thaw cycles because of the temperature changes during storage and transportation. This leads to a redistribution and recrystallisation of the water in the dough. The larger ice crystals then break up the network structure of gluten—shortening the shelf life. Bread quality is also lowered by yeast death and a long wake-up time. Bovine bone collagen hydrolysates have been utilised to address all these issues through its reducing of the number and size of ice crystals—decreasing the freezable water content, reducing water mobility, decreasing the thawing and freezing time, increasing the yeast survival rate, additionally decreasing the hardness, cohesiveness, and chewiness of the frozen dough while increasing its springiness and gumminess. One of the hypothesised mechanisms of action for IBPs inhibition of ice crystal growth is the alignment and binding of the oxygen triad plane of the ${\left(\mathrm{Gly}-X-Y\right)}_{n}$ collagen triplet to the oxygens on the primary and secondary prism faces of the ice [110].

Collagen is also used in the production of environmentally friendly (made from production waste) edible casings (films used for packaging) and coatings (used for preservation). The hydrophobic and hydrophilic collagen side chains tend to collect at the surface of aqueous solutions reducing the surface tension and forming films that surround and are of the same charge as the dispersed phase [111,112]. These films owe their high tensile strength and elasticity to the tropocollagen structure and regular fibril arrangement. Lima et al. investigated edible and semipermeable collagen–galactomannan coatings for mangoes and apples in 2010 and demonstrated their ability to reduce gas transfer rates to extend shelf life [113], while Liu et al. used fish collagen from blue shark fin to create a physical microbial barrier and prevent drip loss preserving and improving the safety of red porgy meat in 2019 [114]. Finally, comparing edible “artificial” collagen sausage casings with “natural” bovine ones Zając and colleagues found that not only were the collagen casings elastic and stronger, but also more acceptable in sensory analysis. To producers, this offers more economical, faster, and more standardised sausage production [115].

The examples presented above are by no means an exhaustive list of the various applications of collagen. For a more comprehensive review the reader is kindly asked to refer to the paper written by Cao and colleagues in 2021 [98].

### 8.5. Gelatine

#### 8.5.1. Structure

Gelatine is obtained by the denaturation of collagen. The method of extraction and the raw material used determine the degree to which collagen is denatured. The obtained unfractionated gelatine is thus a mixture of polypeptide chains of different molecular weights:
Independent, randomly coiled α-chains of $5\times 10^{5}\;g\;{\mathrm{mol}}^{-1}$ to $10\times 10^{5}\;g\;{\mathrm{mol}}^{-1}$;α-chain dimers linked by one or more covalent bonds referred to as β-chains, of molecular weight ranging from $1.2\times 10^{5}\;g\;{\mathrm{mol}}^{-1}$ to $2\times 10^{5}\;g\;{\mathrm{mol}}^{-1}$;α-chain trimers linked by covalent bonds referred to as γ-chains $200\times 10^{3}\;g\;{\mathrm{mol}}^{-1}$ to $400\times 10^{3}\;g\;{\mathrm{mol}}^{-1}$ [116,117,118].


One should note that small differences exist between the native collagen monomer and the gelatine α-chains. One of them, mentioned earlier, is the removal of amide groups from asparagine and glutamine leading to an increase in the aspartic and glutamic acid content during the alcaic pre-treatment. Prolonged liming processes can result in the conversion of arginine to ornithine through the removal of a urea group from the arginine side chain. Finally, cysteine, tyrosine, isoleucine, serine, etc., are present in lower concentrations due to the removal of some of the telopeptide during the denaturation process and its loss in the pre-treatment solution [119].

#### 8.5.2. Properties and Applications

Pure and dry gelatine is available on the market as a colourless or slightly yellow to amber, translucent, tasteless, brittle, and glassy solid. Similar to collagen, it has gelling and water-binding properties making it a particularly useful additive in the food industry. The mechanism by which gelatine gelation occurs, as well as the resulting gel network structure itself is, however, different. Collagen gelation involves molecule and fibril aggregation as a result of ionic strength, pH, and temperature changes. During a lag phase collagen dimers and trimers are nucleated, after which lateral microfibrillar aggregation occurs until equilibrium is reached. This occurs spontaneously when the temperature is raised from 20 to 28 °C. Gelatine gelation, on the other hand, occurs when its solution is cooled below 30 °C and involves the creation of helices, similar to the tropocollagen triple helices. What is more, no equilibrium is reached. The gelation process is thermoreversible in both cases, but melting occurs by lowering the temperature in the case of collagen gels and by increasing the temperature in the case of gelatine gels [112]. The exact mechanism of gelatine gelation has not yet been fully explained. The hypotheses proposed involve small sections of a number of gelatine molecules uniting to form crystallites, creating a 3D network immobilising the liquid and forming a gel. This could be due to hydrogen and van der Waals forces and/or peptides linkages [120]. Perhaps the most obvious food-related applications resulting from this property are jelly desserts, one example being the gelatine-containing American Jell-O^®^ [121]. Gelatine offers creaminess, a “melt-in mouth” experience, and a possibility of fat-reduction [117]. Gelatine is furthermore used as a gelling agent in confectionery, for example in gummy sweets, also providing chewiness and texture [122].

Gelatine is also used in confectionery due to its foaming ability, for example in marshmallow production [123]. Foaming increases with the amount of hydrophobic amino acids such as alanine, valine, isoleucine, leucine, proline, methionine, phenylalanine, tyrosine, and tryptophan, as hydrophobicity is a requirement for adsorption at the air–water interface [124].

The baking industry also benefits from gelatine as a foam-producing material as well as setting agent and stabilising substance in pies, breads, cakes, and even icings [120]. What is more, Yu et al., have investigated the effect of pigskin gelatine on dough properties and bread quality to elucidate the mechanism by which gelatine prevents staling through starch retrogradation and water migration. The retrogradation of amylopectin is, in contrast to that of amylose, a long-term process, which is why it is believed to be the main contributor to the bread staling phenomenon. Water migration and equilibration between crumb and crust is also thought to contribute to the worsening of the bread quality over time. Gelatine appeared to decrease the staling rate of bread in the study conducted by Yu et al. The X-ray diffraction patterns and average values of total crystallinity for crumb suggested that gelatine reduced the water mobility in bread and, consequently, the availability of water for the formation of crystal lattice. Less free water also meant less plasticiser and thus the migration of starch molecules during the retrogradation of amylopectin. Lastly, hydrogen bonding between gelatine and the side chains of amylopectin could have hindered the reassociation of amylose and amylopectin. The 5-day moisture loss rate was the lowest in the gelatine-containing bread suggesting gelatine improved the water-holding capacity of bread in all likelihood due to the gelatine hydroxyl group bonding with, and restricting the movement of, water [125].

Gelatine ability to bind water is used in the meat industry as well and it has found many uses similar to those described for collagen. These include preventing juice loss and providing a good heat transfer medium during the cooking of canned meat products including ham, loaves, frankfurters, wiener-type sausages, and cured, canned pork [117]. To address the lower durability and rates of expansion of kibble resulting from the current tendency towards a higher protein and lower starch content, Manbeck et al. investigated the effect of adding gelatine to the kibble. While increased gelatine strength (bloom) increased product expansion, most probably due to a foaming effect, durability declined with mid and high-bloom gelatine. The authors concluded low bloom gelatine could be used as a binder in pet-food [126]. Just like collagen, gelatine can also be used as a fat replacement rendering lighter meat products. Serdaroğlu et al. [127] showed that gelled emulsion containing gelatine as a partial replacement of beef fat improved cooking yield without negative effects on water holding capacity and emulsion stability.

Gelatine is also used for the production of edible, biodegradable, and bioactive films. The film formation mechanism is similar to that described for collagen. Fakhouri et al. coated red crimson grapes with composite films based on corn starch and gelatine, plasticised with sorbitol. Gelatine increased the mechanical strength, water vapour permeability and thickness of the biofilms. The films improved appearance and lowered the weight loss after 21 days of storage in refrigerated conditions [128]. For a more comprehensive review of gelatine-based composite film applications the reader is kindly asked to refer to [129].

Like collagen, gelatine can also be used as a clarifying agent in beverages, for example in beer, wine, fruit juices, and vinegar [117]. This was demonstrated by Mulyani et al. through successfully using commercial bovine gelatine to precipitate the substances that cause turbidity in apple juice and star fruit juice and reduce the concentration of polyphenols, such as tannins and anthocyanogens. This was explained by the positively charged gelatine residues forming hydrogen bonds with the negatively charged polyphenol and anthocyanogen groups resulting in a complex of compounds insoluble in the fruit juice [130].

Finally, gelatine coacervates complexed with anionic polymers in the form of microcapsules are able to entrap functional components and protect against oxidation or degradation prolonging shelf life. This is described in detail by Gomez et al. [112].

## 9. Natural Rubber

Natural rubber, also known as latex, is a natural additional polymer of a diene monomer called isoprene (C_5_H_8_). It is the only hydrocarbon among so many different types of biopolymers, being nonpolar and thus hydrophobic. Each repeating unit of natural rubber contains one carbon-carbon double bond with *cis* configuration. Natural rubber deteriorates relatively easily via auto-oxidation, due to the high reactivity of allylic hydrogen bonded to the carbon atom adjacent to the double bond. The oxidation is caused by the atmospheric oxygen and ozone. Therefore, antidegradants are added to increase its lifetime. Furthermore, the sulphur vulcanisation of natural rubber leads to the formation of network structure. Natural rubber is highly elastic, which makes it an outstanding raw material for elastomeric devices. The main source of natural rubber is rubber tree, *Hevea brasiliensis*. Due to biosecurity problems alternative sources of this biopolymer, such as guayule and rubber dandelion, are being used more and more often [131]. As a result of the manufacture process, latex products contain added chemicals and proteins from the rubber tree sap. A few of them are known to cause allergic reactions to latex. Importantly, some foods (e.g., avocados and bananas) are known to cross-react with latex. Latex gloves are commonly used by food handlers to prevent pathogen contamination of food. This may result in food contamination with latex proteins in consequence leading to allergic reactions in consumers [132].

## 10. Conclusions and Future Perspectives

The selected polymers presented in the present review constitute by no means an exhaustive list of all the biopolymers important in food-related applications, and the potential regret of a reader who could not find the description of substances such as cellulose, lignin, PHA, chitin, alginates, or hemicellulose can be understood. However, the authors had to make a subjective choice to describe only some most prevalent and safest biopolymers with the widest spectra of food-related applications. A significant group described hereby are these biopolymers, which—as a consequence of the latest progress—are becoming increasingly applied in the production of better active and smart packaging. They improve the quality and shelf life of food products, and at the same time replace fossil-based traditional plastics responsible for pollution, which has become a global concern. Future perspectives of biopolymers’ applications in the food and pharmaceutical industries are attracting increasing interest. However, they need to be examined toxicologically before their widespread administration to human beings, and unequivocal evidence of their safety needs to be obtained. The biopolymer-based supports for enzyme immobilisation have many advantages, such as biocompatibility, biodegradability, high reusability, economic efficiency, and outstanding stability even in drastic environmental conditions, which makes them attractive for the numerous developing areas of biotechnology, chemical syntheses, food industry, and pharmacology. The latest progress in enzyme immobilisation on various carriers and novel immobilisation methods may result in introducing a multifunctional optimised biocatalytic model in the future. Although this paper is focused strongly on structures, properties, and food-related applications of biopolymers, the information presented herein could be useful and helpful to the wider group of scientists working in the fields of tissue engineering, food technology, biotechnology, pharmacology, and medicine.

## Figures and Tables

**Figure 1 polymers-14-01962-f001:**
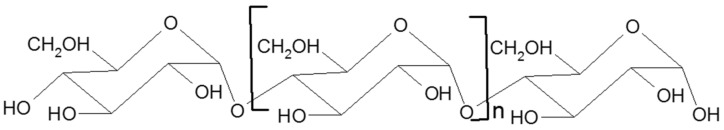
Schematic structure of amylose.

**Figure 2 polymers-14-01962-f002:**
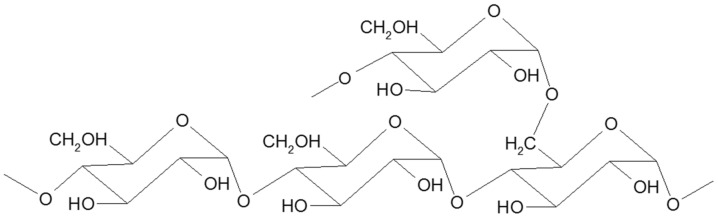
Schematic structure of amylopectin.

**Figure 3 polymers-14-01962-f003:**
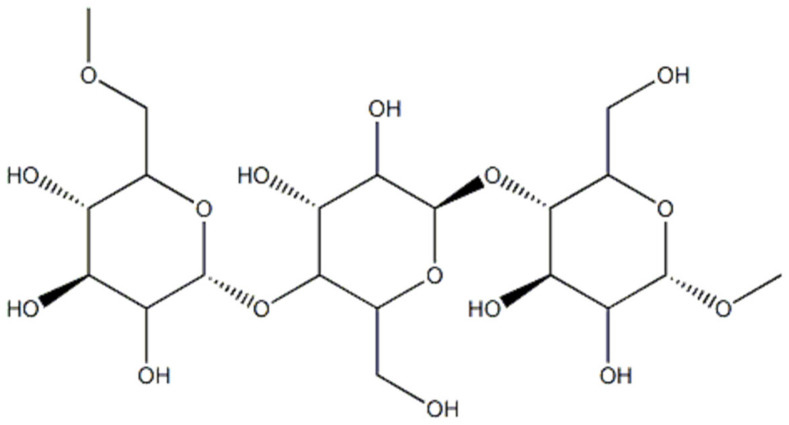
Representation of the pullulan structure.

**Figure 4 polymers-14-01962-f004:**
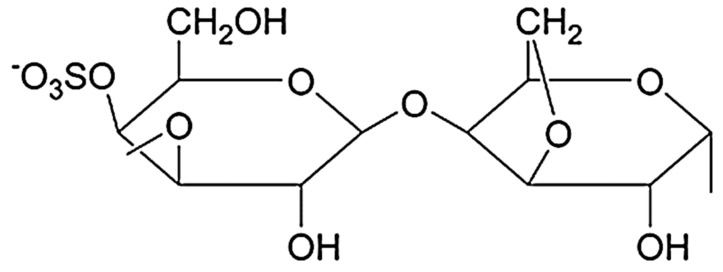
Representative of carrageenan structure (κ–CG).

**Figure 5 polymers-14-01962-f005:**
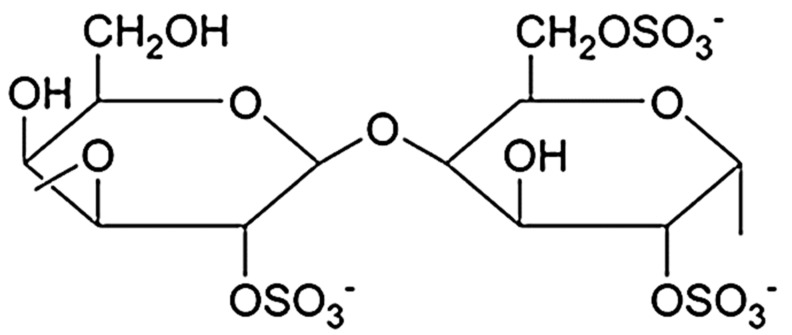
Chemical structure of the repeating unit of λ–CG.

**Figure 6 polymers-14-01962-f006:**
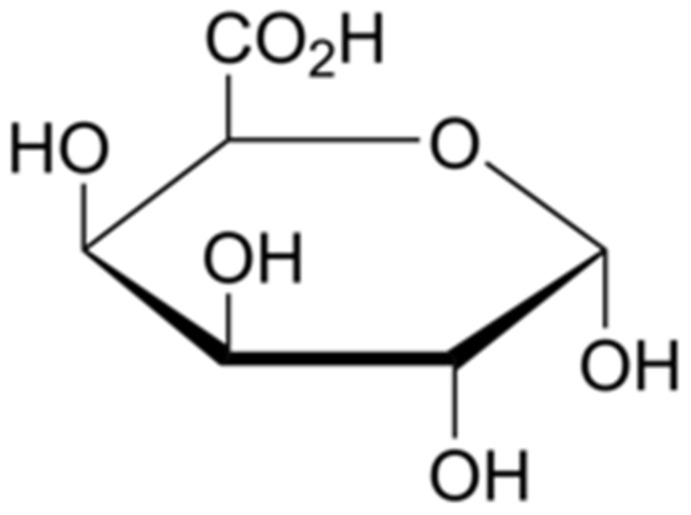
Chemical structure of D-galacturonic acid (GalA).

**Figure 7 polymers-14-01962-f007:**
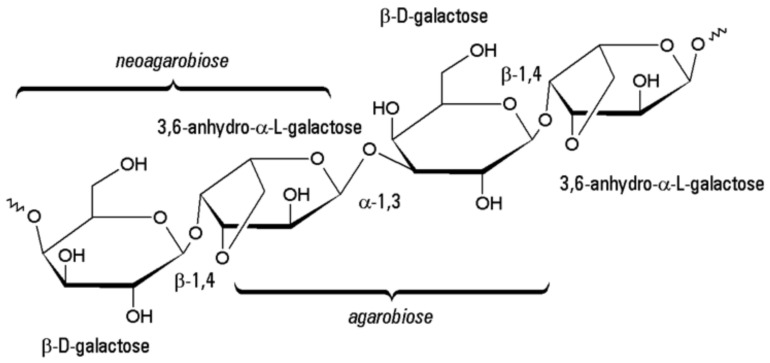
Structure of agar.

**Figure 8 polymers-14-01962-f008:**
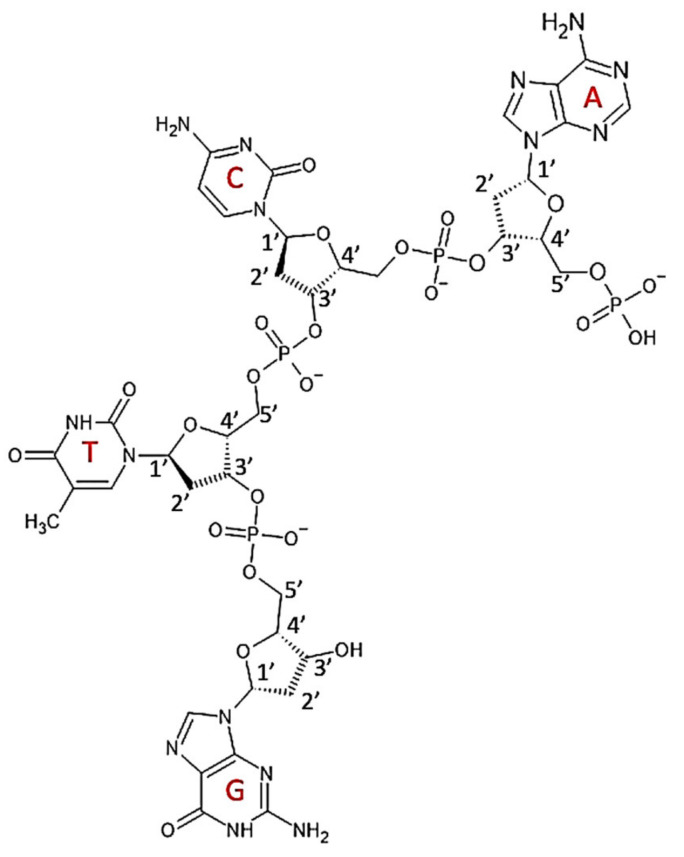
Single-strand DNA fragment drawn using ACD/ChemSketch. Red letter represents a nitrogenous base: A—adenine, C—cytosine, T—thymine, and G—guanine.

**Figure 9 polymers-14-01962-f009:**
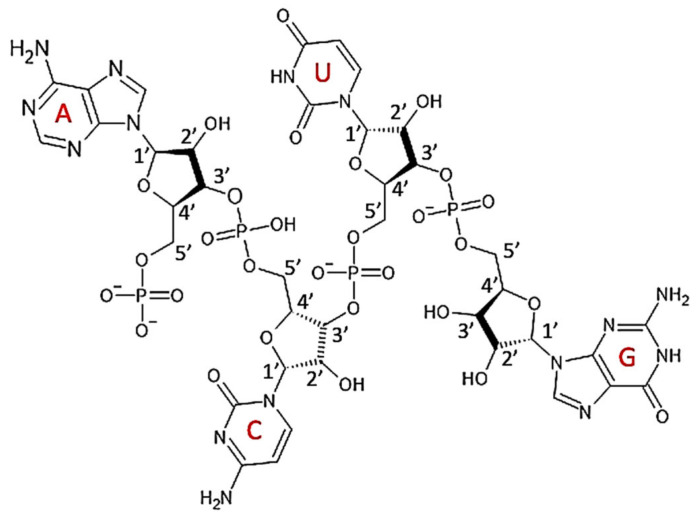
RNA fragment drawn using ACD/ChemSketch. Red letter represents a nitrogenous base: A—adenine, C—cytosine, U—uracil, and G—guanine.

**Figure 10 polymers-14-01962-f010:**
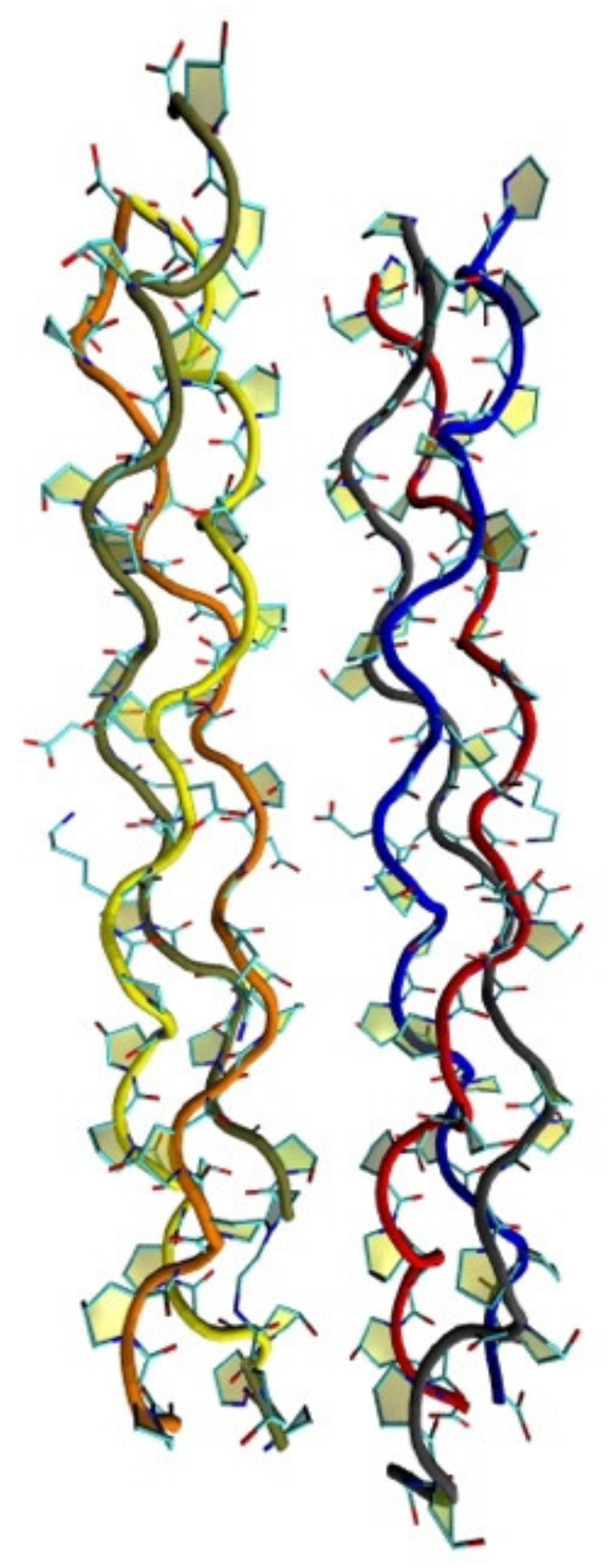
Tropocollagen structure.

**Table 1 polymers-14-01962-t001:** The comparison of structure, properties and applications of unmodified and UA-treated starch.

Treatment	Structure	Property	Application
Unmodified	high amylose content	high mechanical strength	strong films
	or		
	high amylopectin content	low mechanical strength, unstable under stress, at high temp. and pH	films require addition of plasticisers
UA	more amylose, less amylopectin	higher strength, higher swelling, better solubility	stronger, more transparent films
	de-polymerisation	lower viscosity	
	more RDS		limited nutritional application

**Table 2 polymers-14-01962-t002:** The comparison of structure, properties and applications of unmodified and modified carrageenans.

Treatment	Structure	Property	Application
Unmodified	basic	water solubility, viscosity, and gelling ability depending on the number of sulphate groups	edible films, active food packaging
	κ-CG	antimicrobial activity, high water uptake	composite superabsorbent gel
	κ-CG, ι-CG	high antioxidant activity	edible films, active food packaging, functional food
	κ-CG, ι-CG, λ-CG	high antiviral activity	antiviral drugs
Carboxymethylation	additional carboxymethyl groups	higher water solubility	drug delivery/controlled release

**Table 3 polymers-14-01962-t003:** Examples of enzymes and their most up-to-date food-related applications.

Enzyme [Ref.]	Substrate(s) and Reaction	Food-Related Application
laccase [75]	oxidation of phenols, carbohydrates, unsaturated fatty acids and thiol-containing proteins with a concomitant reduction of oxygen to water	improving the volume, texture, flavour and freshness of bakery products; improving the dough elastic properties of gluten-free flour; stabilising agent preventing the formation of sediments, haze, turbidity; antioxidant synthesis of novel antioxidants for food industry; cross-linking agent used in, e.g., gel or film formation
naringinase and α-L-rhamnosidase [76]	conversion of naringin into bitterness products	debittering agents in citrus fruit juices; increases the shelf life of juice
pectinase [77]	degradation of pectic molecules	clarification of various fruit juices
β-galactosidase *Tt*bGal1 [78]	hydrolysis of lactose and formation of galactooligosaccharides via transgalactosylation	lactose-reduced or lactose-free products; probiotic properties
alcalase and flavourzyme [79]	hydrolysis of whole whey proteins	hydrolysates with less sulfhydryl groups; enhanced antioxidant capacity, natural preservatives
the thermophilic esterase EST2 [80]	lipolysis of triglycerides	increase in the production of short- and medium-chain fatty acids via lipolysis consequently leading to volatile compounds formation; enhancement of cheese flavour and reduction in the time of ripening
L-asparaginase from *Melioribacter roseus* [81]	conversion of L-asparagine into L-aspartic acid and ammonia	prevention of the acrylamide formation from the conversion of asparagine during some food processing
lactose oxidase [82]	oxidation of lactose to lactobionic acid with a concomitant reduction of oxygen to water	control of the outgrowth of *L. monocytogenes* in milk, cheese and other dairy products increasing their safety
novel cold-adapted calcium-activated transglutaminase [83]	cross-linking of lysine and glutamine residues of various polypeptides, e.g., casein, collagen and gelatine	increase in the mechanical stability of meat; can be used in treatments requiring low temperatures

## Data Availability

The data presented in this study are available on request from the corresponding author.

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
