# Peer review of "The Influence of the Structure of Selected Polymers on Their Properties and Food-Related Applications"

_polymers, 2022, doi:10.3390/polym14101962_

Round 1

Reviewer 1 Report

The submitted review article contains already well-known information available in different books, chapters and review articles. No innovation comes from the organization or systematic presentation of the information either. In fact, the organization of the information is not at all standardized, with subsection titles that differ among biopolymers, very different figures styles (monomers in some cases, polymers in others; different font type and size, etc.) that have been “cut and paste” from other sources (authors are urged to draw their own structures), sections lengths that differ among biopolymers, and so on.

On the other hand, in sections devoted to biopolymers applications, it does not come clear how the examples of biopolymers uses provided have been chosen, since they seem very particular ones and no exhaustive revision of recent literature was provided.

Moreover, from the title and the introduction, the review is expected to be focused on food-related applications. However, for some biopolymers long descriptions of for example medical uses are provided (e.g. carrageenan). Besides, the aim of the contribution claimed in the abstract (i.e. “… this paper highlights also how the method of extraction and treatment affects the structure, properties and hence applications of polysaccharides…”) was not fulfilled (not even included) for some of the biopolymers described (e.g. pullulan).

English use is also poor and extensive revision by a native speaker is required.

Other details:

  • Many typos were found. E.g. Figure 1 caption: amylase instead of amylose.
  • Conventional biopolymers extraction should be described prior to mentioning the benefits derived from adding ultrasonic treatment.
  • Lines 172-173. Starch is already a biopolymer. Please revise.
  • When referring to each biopolymer application, links to recognized reviews on the topic (and not just to specific articles) should be included to provide readers with a higher number of examples and a more exhaustive revision of literature (at least by referring to other specific review articles).
  • Section 4.3. Please avoid repetitions (e.g. applications in films production).
  • Collagen should have been included in the section referring to proteins.
  • Cellulose, lignin and hemicellulose, with applications for example as polymeric films reinforcements, and/or fillers with antioxidant and UV blocking activity should have been described given their use as food packaging.

Reviewer 2 Report

Review article titled (The Influence of the Structure of Selected Polymers on their Properties and Food-Related Applications) by Koczoń et al. is a well organized and well written article. I liked the way the authors organized their work and I have some comments for imptovement before publishing:

1- kindly revise all abbreviations and identification at the first appearance.

2- Numbering should be revised and start "1" at Starch

3- The review in general is too long and needs to be shortened to be more concrete

4- Number of references is huge, can the authors reduce the number?

Reviewer 3 Report

The present review paper presents the structure-food-related applications of several polymers obtained from natural sources, such as starch, proteins, DNA, collagen, gelatine, pullulan, carrageenans, pectins, agar, and natural rubbers. The manuscript contains excessive information related to each of these polymers and pretends to make a correlation between the structure and food applications. The manuscript has severe flaws needing attention before publication. First of all, a review paper must present and scope and goal. Here, it is hard to find this focus. The abstract is not well-written and does not make a clear point about the manuscript's importance, scope, and content. The novelty of the manuscript is not carefully explained. The introductions lack a coherent structure. There are no cited references; there is no background or state of the art and problems as main points. Besides, the chemical structures are presented in low quality. Some sugar rings are in a chair structure, and some are in six-carbon rings. That does not make sense. The figures' resolution is too low. I do not understand why presenting excessive information in each section. It is possible to make schemes for extraction procedures, tables compiling information, trying to avoid excessive details that make the reader feel tired. Several typos are highlighted in the attached version. Finally, some references are cited several times in continuous paragraphs in several sections of the manuscript. That makes me feel unconfident about the length of the information extracted from those references.

Also, several citations to already existing review papers make me suppose the lack of novelty about the present work. Some of the keywords are too general (structure, properties, applications), not keywords in the manuscript. Sometimes, finding the main point in the history of a section is complex. For example, in the starch section of properties and applications, lines 137-138 suddenly present starch nanoparticles, which lack cohesion with the previous information. Also, ¿what were the criteria for choosing these polymers? What about polyhydroxyalkanoates (PHA), gums, chitin, alginates, etc.? The manuscript needs better organization. Sections must have subsections to emphasize applications such as encapsulants, catalysts in food-processing, coatings, etc. I suggest a careful review of the manuscript. A comprehensive review, or a literature review, needs a concise organization to understand the content easily. I believe that the manuscript content is relevant for the food industry but needs attention in all the made suggestions. Use schemes, figures, and tables for easy comprehension. The attached version contains some examples, but it is not an exhaustive list. 

Round 2

Reviewer 1 Report

Some of my comments have been considered, but the most important arguments against publication are still there.

Author Response

The authors are very grateful for Reviewer’s efforts in preparing of the review report.

Reviewer 2 Report

I would suggest accepting the current form of the paper.

Author Response

The authors would like to express their gratitude for Reviewer’s valuable report.

Reviewer 3 Report

I appreciate the effort from the authors for the manuscript's improvement. It is a much better version than the previous. Also, several suggestions were conducted correctly. Related to point 8: "Some sugar rings are in a chair structure, and some are in six-carbon rings," I followed the recommended reference and could not get the author's point. The table mentioned talks about biopolymers and cited some concerns. However, that is not related to my point of "homogenizing" the chair or ring structures for carbohydrates. It was just a suggestion, but the fact is also argued, and I agree with the analysis. 
